# Comparison of dosimetric effects of MLC positional errors on VMAT and IMRT plans for SBRT radiotherapy in non-small cell lung cancer

Jia Deng[1,2]*, Yun Huang[3], Xiangyang Wu[1], Ye Hong[4], Yaolin Zhao[2]

1 Department of Radiation Oncology, Shaanxi Provincial Cancer Hospital, Xi'an, Shaanxi, People's Republic of China, 2 School of Nuclear Science and Technology, Xi'an Jiaotong University, Xi'an, Shaanxi, People's Republic of China, 3 Department of Radiation Oncology, Xianyang Central Hospital, Xi'an, Shaanxi, People's Republic of China, 4 Center of Digestive Endoscopy, Shaanxi Provincial Cancer Hospital, Xi'an, Shaanxi, People's Republic of China

* dengjia92@yeah.net

**Data Availability Statement:** All minimal dataset files are available from the figshare database (accession number(s) https://doi.org/10.6084/m9.figshare.21366744.v1).

## Abstract

The positional accuracy of multi-leaf collimators (MLC) is important in stereotactic body radiotherapy (SBRT). The aim of this study was to investigate the impact between MLC positional error and dosimetry of volume intensity modulated (VMAT) and general intensity modulated (IMRT) plans for non-small cell lung cancer (NSCLC). Fifteen patients with NSCLC were selected to design the 360 SBRT-VMAT plans and the 360 SBRT-IMRT error plans. The DICOM files for these treatment plans were imported into a proprietary computer program that introduced delivery errors. Random and systematic MLC position (0.1, 0.2, 0.5, 1.0, 1.5, and 2.0 mm) errors were introduced. The systematic errors were shift errors (caused by gravity), opening errors, and closing errors. The CI, GI, $d_{2cm}$ and generalized equivalent uniform dose (gEUD) were calculated for the original plan and all treatment plans, accounting for the errors. Dose sensitivity was calculated using linear regression for MLC position errors. The random MLC errors were relatively insignificant. MLC shift, opening, and closing errors had a significant effect on the dose distribution of the SBRT plan. VMAT was more significant than IMRT. To ensure that the gEUD variation of PTV is controlled within 2%, the shift error, opening error, and closing error of IMRT should be less than 2.4 mm, 1.15 mm, and 0.97 mm, respectively. For VMAT, the shift error, opening error, and closing error should be less than 0.95 mm, 0.32 mm, and 0.38 mm, respectively. The dose sensitivity results obtained in this study can be used as a guide for patient-based quality assurance efforts. The position error of the MLC system had a significant impact on the gEUD of the SBRT technology. The MLC systematic error has a greater dosimetric impact on the VMAT plan than on the IMRT plan for SBRT, which should be carefully monitored.

**Funding:** JD was supported by grants from the Health research Program of Shaanxi Provincial Health Commission (No. 2022D037). The funders had no role in study design, data collection and analysis, decision to publish, or preparation of the manuscript.

**Competing interests:** The authors have declared that no competing interests exist.

## Introduction

Non-small cell lung cancer (NSCLC) is the most common form of lung cancer and is associated with increased morbidity and mortality [1–3]. Stereotactic body radiotherapy (SBRT) has become the standard treatment for patients with medically inoperable early-stage NSCLC. Compared with conventionally fractionated radiotherapy, SBRT allows larger doses to be delivered in a small number of fractions, resulting in a higher biologically effective dose [4]. To limit and minimize radiation-induced damage to normal tissue, SRBT requires the concentration of high doses to the planned target volume (PTV) and a large dose fall-off outside the PTV. Linac-based SBRT can be mainly performed with intensity-modulated radiotherapy (IMRT) and volumetric-modulated arc therapy (VMAT). Modulated dose distribution in IMRT plans can be achieved with a multi-leaf collimator (MLC), in which the leaves are in motion during irradiation. VMAT delivers a dose with precise synchronization of the gantry rotation speed, MLC motion, and dose rate [5–7]. Previous studies have found that the conformity and uncertainty of SBRT dose delivery are more dependent on the position of the MLC leaf than conventionally fractionated radiotherapy, which could lead to more serious complications [8–10]. Therefore, it is necessary to investigate the dosimetric effects of MLC leaf position errors in both IMRT and VMAT for SBRT. The aim of this study was to investigate the effects of systematic and random MLC leaf position errors on lung SBRT with IMRT/VMAT in patients with NSCLC. We investigated the differences in dose sensitivity between IMRT and VMAT for four types of MLC errors and sought to determine their benefits and limitations.

## Methods and materials

### Patient selection

Fifteen patients clinically diagnosed with stage I–II central lung cancer were selected for this study. They were ten men and five women, aged between 45 and 76 years, with a planned target volume between 3.6 and 13.4 cm3. Ten patients had squamous cell carcinoma and five had adenocarcinoma. The prescribed dose for lung SBRT was 50 Gy to the PTV, and the limited dose of OARs was considered according to RTOG protocols. All data were obtained after the ethics committee (Medical ethics committee of Shaanxi Provincial Cancer Hospital) approval, informed consent waiver, and verbal informed consent for the study was obtained.

### Treatment planning

Two types of SBRT plans, IMRT and VMAT, were prepared for each of the fifteen patients. For the VMAT plans, one arc was used with a gantry range of 179˚ to 181˚ with the collimator set to 10˚, and the other was delivered with a collimator angle of -10˚ and a gantry rotation between 181˚ and 179˚. Seven fields were set up in the IMRT plans. The gantry angles of the fields were 0˚, 52˚, 104˚, 156˚, 208˚, 260˚, and 312˚ with a collimator angle of 10˚. Under the same dose optimization conditions, the IMRT and VMAT plans were used for each case.

The plans were created using Eclipse TPS version 13.5 with a photon optimizer for Photon Optimizer (PO) and an Anisotropic Analytical Algorithm (AAA) for dose distribution with a dose calculation grid of 1.0 mm. All plans were reproduced on a TrueBeam (Varian)equipped with 60 pairs of leaves, which had 40 inner- and 20 outer-leaf pairs with a width of 0.5 and 1 cm, respectively.

### Simulation of MLC leaf position errors

Three types of systematic and one type of random MLC position errors were introduced into the original VMAT and IMRT plans of NSCLC for error magnitudes of 0.1, 0.2, 0.5, 1, 1.5, and

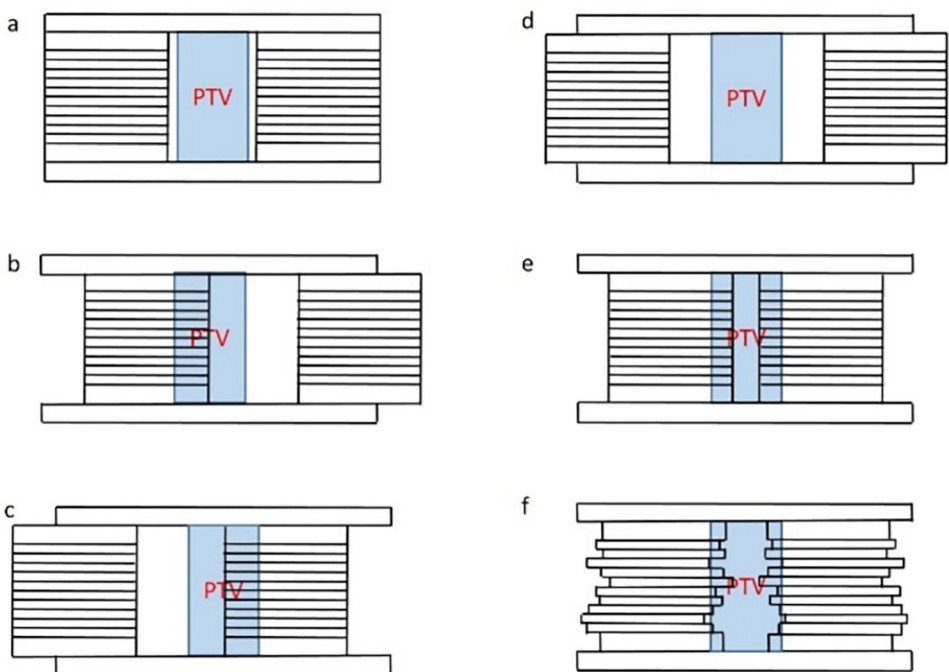

**Fig 1. MLC position errors are introduced to each SBRT plan.** (a) baseline plan; (b)(c) Type1, shift error; (d)Type2, opening error; (e)Type3, closing error; (f)Type4, random error.

2 mm. Fig 1, shows a graph illustrating the different types of MLC position errors in this study. Type 1 errors were shift errors in which both MLC banks moved to the left or the right by adding the same error magnitude to each leaf position, while the subfield magnitude remained unchanged. when the gantry is from -179 to 0 degrees, the MLC bank shift to the left, as shown in Fig 1(B), and when the gantry from 0 to 179 degrees, the MLC bank shift to the right, as shown in Fig 1(C) [11]. Considering the influence of gravity on the MLC positions, the direction of the shift error was defined as a function of the gantry angle. Type 2 and type 3 errors were opening and closing errors, and both MLC banks moved in opposite directions with the same error magnitude, increasing and decreasing the MLC leaf gap, respectively. Type 4 errors were simulated random errors that were introduced by adding or subtracting random errors determined by sampling a Gaussian function centered on zero, with a standard deviation equal to the error magnitude. To avoid potential errors and unintentional duplication, we generate a new Gaussian random distribution for each error plan, so that the same errors are not added to different exposure fields and plans.

Plans without introduced errors were referred to as "Baseline" plans. First, the fifteen "Baseline" treatment plans were exported from the TPS and then imported into an in-house program written in MATLAB. The in-house program was used to modify the MLC leaf positions with simulated errors. Then, all error-related treatment plans were imported into the TPS for dose recalculation. If an error resulted in a negative leaf gap, the MLC positions for the gap were re-adjusted to zero.

## Plan evaluation

To evaluate dosimetric differences between the simulated plans and the "baseline" plans, routine plan evaluation methods were used, including isodose distributions and DVH. All plans

were evaluated by conformity index(CI), gradient index(GI), $D_{2cm}$, and generalized Equivalent Uniform Dose (gEUD). The CI reflected the conformity of the shape and size of the isodose envelope to the PTV, the CI represented the conformity of a reference isodose to the PTV, defined as:

$$\text{CI} = \frac{V_{(P,R)}}{V_P} \times \frac{V_{(P,R)}}{V_R} \tag{1}$$

where $V_{(P,R)}$ is the PTV volume covered by the prescribed isodose surface, $V_R$ is the volume covered by the prescribed isodose surface, and $V_P$ is the PTV volume. The GI was defined as the degree of dose drop-off outside the PTV. GI was calculated using the following equation:

$$\text{GI} = \frac{R_{50\%}}{R_{100\%}} \tag{2}$$

where $R_{50\%}$ is the ratio of the 50% prescription isodose volume to the PTV and $R_{100\%}$ is the ratio of the 100% prescription isodose volume to the PTV.

$D_{2cm}$ represents the ratio of the maximum dose to the prescribed dose 2 cm from the PTV in any direction.

gEUD was used to summarize the total DVH in a single metric, defined as the dose that would have the same biological effect if delivered uniformly to the entire volume. gEUD can be expressed as follows [12,13]:

$$gEUD = \left(\sum_i^m v_i D_i^a / \sum_i^m v_i\right)^{1/a} \tag{3}$$

where $m$ is the number of voxels in the volume of interest and $D_i$ and $v_i$ are the dose and volume in the i-th voxel, respectively. $a$ is the tumor-or normal tissue-specific parameter describing the dose–volume effect, which can be obtained from clinical outcome data. According to the research results, the value of $a$ was set to -20 for PTV, 20 for spinal cord, and 1 for both the lungs and heart [14]. In addition, the plans were analyzed to identify the relative percentage change in gEUD for each error magnitude, expressed by $\Delta gEUD_{Error}$:

$$\triangle gEUD_{Error} = \frac{gEUD_{Error} - \text{gEUD}_{Baseline}}{\text{gEUD}_{Baseline}} \times 100\% \tag{4}$$

where the subscript "Error" indicates the plan being evaluated.

## Results

### Statistical analysis

All statistical analyses were performed with SPSS 20.0, and a paired t-test was used to compare the two groups. Experimental data are expressed as mean ± standard deviation, and $P<0.05$ was considered statistically significant in this study.

### DVH comparison between IMRT and VMAT

Fig 2 shows A sample dose–volume histogram (DVH) of the PTV, left lung, lung, heart, and spinal cord for the baseline plan and different simulation plans with an MLC position error of 2 mm for each case. It can be seen from the diagram that systematic MLC opening and closing errors can cause noticeable dosimetric changes in the PTV and OARs. In addition, IMRT plans are less sensitive than VMAT plans to dose deviations caused by the same type of MLC error.

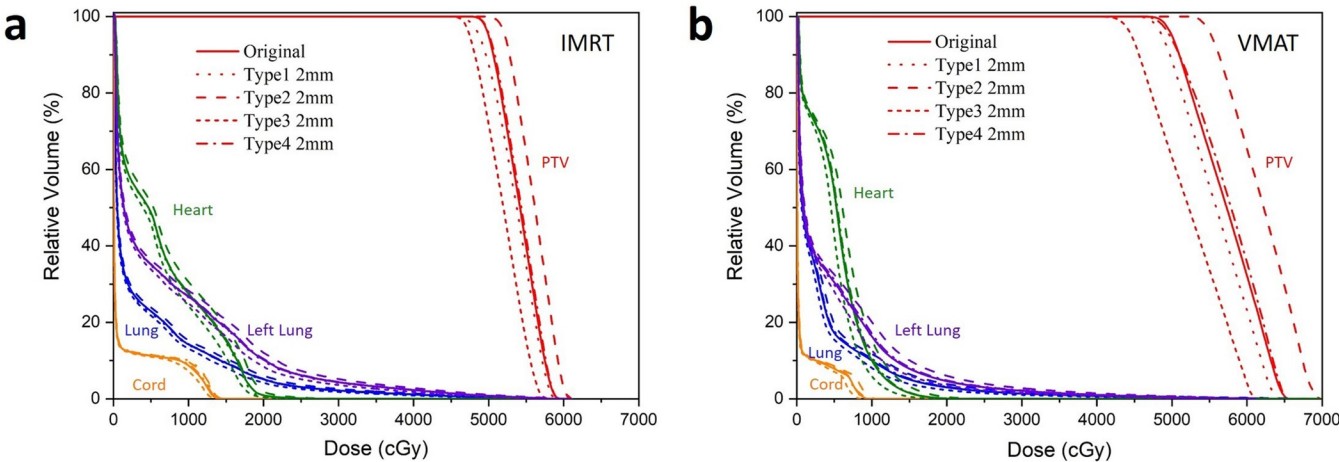

**Fig 2. A sample dose–volume histogram of patient 5.** This figure shows A sample DVH of the PTV, left lung, lung, heart, and spinal cord for the baseline plan and different simulation plans with an MLC position error of 2 mm for each case. 2(a) IMRT plans; 2(b) VMAT plans.

## Dosimetric parameters comparison of PTV between IMRT and VMAT

The results of the percentage changes for CI, GI, and $D_{2cm}$ are shown in Fig 3 for all four types of MLC position errors. Fig 3(A) shows that the CI decreased with increasing MLC error. The changes in the closing error were the largest, followed by the opening error, and the random error was minimal. In addition, IMRT plans are less sensitive than VMAT plans to dose deviations caused by the same type of MLC error.

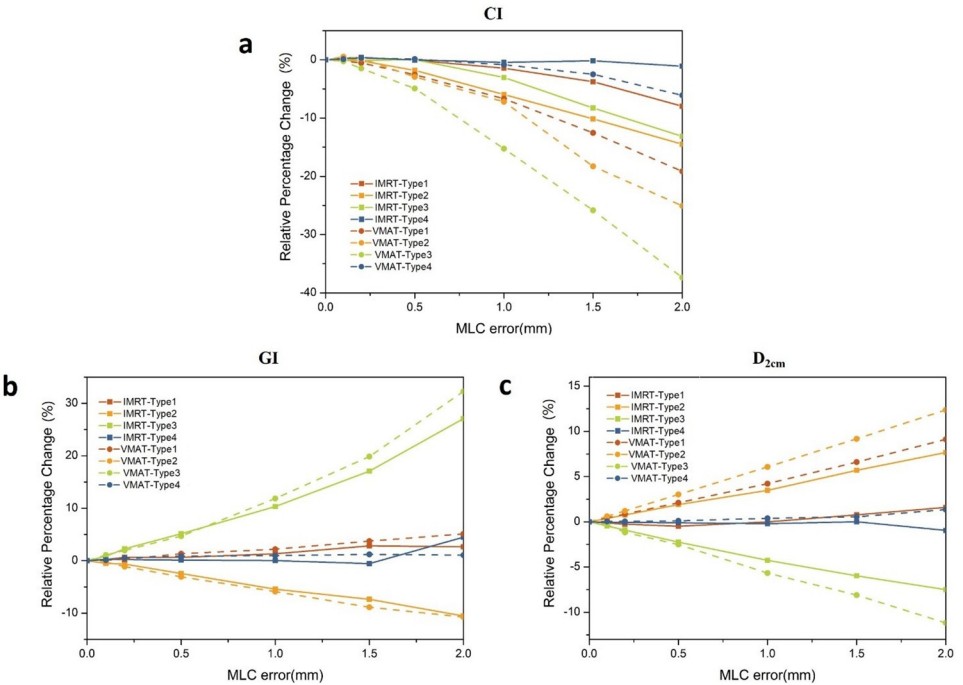

**Fig 3. The percentage changes for CI, GI, and D2cm for all types of MLC position errors in IMRT plans and VMAT plans.** 3(a) CI decreased with increasing MLC error; 3(b) the relative percentage change of GI compared to each type of MLC error; 3(c) the relative percentage change of $D_{2cm}$ compared to each type of MLC error.

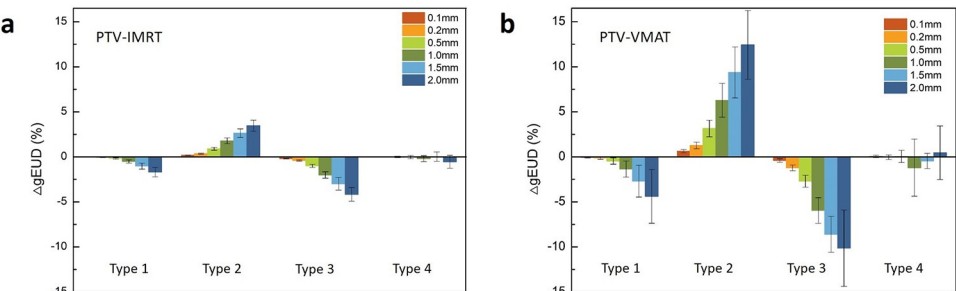

**Fig 4. The gEUD variation in the PTV caused by MLC errors.** 4(a) The gEUD variation in IMRT plans. 4(b) The gEUD variation in VMAT plans.

Fig 3(B) shows the relative percentage change of GI compared to each type of MLC error. It can be seen that the random error had no obvious effect on the GI of IMRT and VMAT plans. The largest change in CI was observed when opening the MLC leaves, which showed a rapid dose fall-off.

As shown in Fig 3(C), $D_{2cm}$ is positively correlated with the opening and shifting errors and decreases with the increase of the closing error. For the type of the same error, the $D_{2cm}$ values for the IMRT plans were slightly lower than those for the VMAT plans.

Fig 4 shows the gEUD variation in the PTV caused by MLC errors. As shown in Fig 4, IMRT plans are less sensitive than VMAT plans to dose deviations caused by the same type of MLC errors. Systematic errors have a greater impact on the PTV gEUD variation than random errors. For an errors up to 2 mm, the changes in PTV gEUD of IMRT and VMAT plans caused by opening errors were 3.27% and 8.11%, respectively, caused by closing errors of -4.17% and -8.14%, respectively, which were caused by a shift error of -1.66% and—4.38% respectively. The influence of the random error is negligible.

## Dosimetric parameters comparison of OAR between IMRT and VMAT

Tables 1–4 summarize the gEUD variation in the OAR of the IMRT and VMAT plans. For the OARs, the random error had a negligible effect on the gEUD. In contrast, the opening and closing errors cause a large change in the gEUD. The biological dose changes caused by shift errors had a significant effect on the whole lung for the IMRT plans and on the heart for the VMAT plans. The biological dose changes for the OARs VMAT plans were significantly higher than those for the IMRT plans when opening or closing the MLC leaves.

## Comparison of dose sensitivity between IMRT and VMAT

Table 5 lists the dose sensitivity values of gEUD for the PTV and the OARs for the four types of MLC errors. The results showed that among the four types of MLC errors, the opening and closing dose sensitivities were the highest. The IMRT plans are seen to exhibit lower sensitivity than VMAT plans. For opening errors, the percentage changes in gEUD of PTV for IMRT and VMAT plans were -2.0623 and-5.3143%/mm, respectively. The dosimetric effects of closing errors are opposite to those of opening errors, and the gEUD sensitivity for the PTV was found to be 1.737 and 6.222%/mm for IMRT and VMAT plans, respectively. The shift error had a small effect on the gEUD, and the gEUD sensitivity values of PTV for IMRT and VMAT plans were -0.819 and -2.136%/mm, respectively. Random errors changed the gEUD insignificantly within 2 mm. For IMRT plans, a gEUD deviation in the PTV of less than 2% ensures that the deviations in shift, opening, and closing errors are less than 2.4, 1.15, and 0.97 mm, respectively. For VMAT plans, to keep the gEUD deviation in the PTV within 2%, it is

**Table 1.  The gEUD variation in the OAR of the IMRT and VMAT plan for shift error.**

| Error | Plan type | Cord (%) | Heart (%) | Lung (%) | Left Lung (%) |
|---|---|---|---|---|---|
| 0.1mm | IMRT | 0.114±0.409 | -0.06±0.132 | -0.083±0.040 | -0.07±0.041 |
| | VMAT | -0.018±0.101 | -0.074±0.098 | -0.005±0.039 | 0.003±0.044 |
| | P | 0.031 | 0.162 | <0.01 | <0.01 |
| 0.2mm | IMRT | 0.219±0.809 | -0.137±0.27 | -0.166±0.08 | -0.151±0.079 |
| | VMAT | -0.036±0.199 | -0.146±0.196 | -0.007±0.077 | 0.004±0.088 |
| | P | 0.031 | 0.163 | <0.01 | <0.01 |
| 0.5mm | IMRT | 0.483±1.935 | -0.306±0.649 | -0.428±0.194 | -0.384±0.193 |
| | VMAT | -0.087±0.487 | -0.359±0.492 | -0.017±0.194 | -0.016±0.188 |
| | P | 0.029 | 0.163 | <0.01 | <0.01 |
| 1.0mm | IMRT | 0.836±3.803 | -0.661±1.279 | -0.911±0.383 | -0.801±0.386 |
| | VMAT | -0.154±0.936 | -0.707±0.982 | -0.041±0.388 | 0.002±0.444 |
| | P | 0.028 | 0.165 | <0.01 | <0.01 |
| 1.5mm | IMRT | 1.113±5.686 | -1.072±1.892 | -1.45±0.575 | -1.275±0.587 |
| | VMAT | -0.201±1.342 | -1.057±1.456 | -0.081±0.581 | -0.02±0.673 |
| | P | 0.026 | 0.169 | <0.01 | <0.01 |
| 2.0mm | IMRT | 1.268±7.528 | -1.58±2.428 | -2.066±0.776 | -1.664±0.985 |
| | VMAT | -0.215±1.697 | -1.39±1.938 | -0.137±0.774 | -0.06±0.908 |
| | P | 0.025 | 0.175 | <0.01 | <0.01 |

necessary to ensure that the deviations in the shift, opening, and closing errors are less than 0.95, 0.32, and 0.38 mm, respectively.

## Discussion

The accuracy of the MLC position is a critical factor for the quality of treatment in clinical inverse intensity-modulated radiotherapy. Especially in fractionally high-dose radiotherapy,

**Table 2.  The gEUD variation in the OAR of the IMRT and VMAT plan for opening error.**

| Error | Plan type | Cord (%) | Heart (%) | Lung (%) | Left Lung (%) |
|---|---|---|---|---|---|
| 0.1mm | IMRT | 0.307±0.233 | 0.491±0.065 | 0.408±0.053 | 0.404±0.055 |
| | VMAT | 0.645±0.153 | 0.774±0.151 | 0.742±0.157 | 0.753±0.16 |
| | P | 0.033 | 0.169 | <0.01 | <0.01 |
| 0.2mm | IMRT | 0.627±0.49 | 0.985±0.132 | 0.818±0.107 | 0.81±0.11 |
| | VMAT | 1.301±0.323 | 1.553±0.297 | 1.545±0.272 | 1.507±0.32 |
| | P | 0.074 | 0.176 | <0.01 | <0.01 |
| 0.5mm | IMRT | 1.518±1.133 | 2.471±0.333 | 2.047±0.267 | 2.029±0.276 |
| | VMAT | 3.227±0.765 | 3.877±0.751 | 3.712±0.784 | 3.769±0.801 |
| | P | 0.037 | 0.197 | <0.01 | <0.01 |
| 1.0mm | IMRT | 3.098±2.412 | 4.943±0.669 | 4.094±0.535 | 4.057±0.552 |
| | VMAT | 6.466±1.533 | 7.771±1.499 | 7.428±1.572 | 7.542±1.607 |
| | P | 0.043 | 0.241 | <0.01 | <0.01 |
| 1.5mm | IMRT | 4.618±3.595 | 7.418±1.011 | 6.139±0.804 | 6.083±0.829 |
| | VMAT | 9.718±2.31 | 11.667±2.258 | 11.152±2.365 | 11.32±2.419 |
| | P | 0.051 | 0.294 | <0.01 | <0.01 |
| 2.0mm | IMRT | 6.140±4.809 | 9.891±1.351 | 8.182±1.072 | 8.108±1.106 |
| | VMAT | 12.985±3.101 | 15.574±3.021 | 14.879±3.163 | 15.102±3.235 |
| | P | 0.06 | 0.358 | <0.01 | <0.01 |

**Table 3. The gEUD variation in the OAR of the IMRT and VMAT plan for closing error.**

| Error | Plan type | Cord (%) | Heart (%) | Lung (%) | Left Lung (%) |
|---|---|---|---|---|---|
| 0.1mm | IMRT | -0.351±0.287 | -0.49±0.066 | -0.414±0.054 | -0.41±0.055 |
|  | VMAT | -0.414±0.159 | -0.423±0.214 | -0.444±0.124 | -0.464±0.129 |
|  | P | 0.032 | 0.162 | <0.01 | <0.01 |
| 0.2mm | IMRT | -0.7±0.572 | -0.974±0.13 | -0.824±0.107 | -0.817±0.109 |
|  | VMAT | -1.248±0.247 | -1.449±0.21 | -1.391±0.237 | -1.4±0.24 |
|  | P | 0.031 | 0.152 | <0.01 | <0.01 |
| 0.5mm | IMRT | -1.774±1.43 | -2.428±0.32 | -2.053±0.271 | -2.039±0.27 |
|  | VMAT | -2.692±0.548 | -3.153±0.51 | -3.037±0.541 | -3.071±0.551 |
|  | P | 0.031 | 0.145 | <0.01 | <0.01 |
| 1.0mm | IMRT | -3.538±2.809 | -4.846±0.636 | -4.104±0.534 | -4.072±0.537 |
|  | VMAT | -6.092±1.105 | -7.025±0.942 | -6.769±1.076 | -6.794±1.082 |
|  | P | 0.027 | 0.118 | <0.01 | <0.01 |
| 1.5mm | IMRT | -5.236±4.087 | -7.26±0.946 | -6.155±0.795 | -6.106±0.804 |
|  | VMAT | -8.679±1.524 | -10.037±1.351 | -9.667±1.513 | -9.712±1.527 |
|  | P | 0.026 | 0.108 | <0.01 | <0.01 |
| 2.0mm | IMRT | -6.955±5.341 | -9.664±1.251 | -8.204±1.054 | -8.138±1.067 |
|  | VMAT | -11.93±1.975 | -11.902±4.772 | -13.211±1.948 | -13.23±1.951 |
|  | P | 0.024 | 0.1 | <0.01 | <0.01 |

referred to as SBRT, the MLC leaf position error leads to a change in the planned dose distribution. Therefore, the accuracy of the MLC position can be ensured during the treatment process, to take advantage of the planned intensity-modulated radiotherapy.

The accuracy of MLC leaf positioning is highly dependent on leaf position repeatability precision, calibration, motor age, and gravity effects, among other factors [15]. A deviation in the MLC leaf position directly results in a deviation in target and OAR dose [9]. ICRU 142

**Table 4. The gEUD variation in the OAR of the IMRT and VMAT plan for random error.**

| Error | Plan type | Cord (%) | Heart (%) | Lung (%) | Left Lung (%) |
|---|---|---|---|---|---|
| 0.1mm | IMRT | -0.01±0.113 | -0.003±0.05 | 0.005±0.045 | 0.006±0.048 |
|  | VMAT | 0.049±0.101 | 0.048±0.135 | 0.063±0.12 | 0.067±0.113 |
|  | P | 0.032 | 0.176 | <0.01 | <0.01 |
| 0.2mm | IMRT | 2.113±6.965 | -0.058±0.15 | -0.07±0.115 | -0.068±0.118 |
|  | VMAT | 0.07±0.233 | -0.088±0.329 | -0.076±0.212 | -0.027±0.249 |
|  | P | 0.033 | 0.162 | <0.01 | <0.01 |
| 0.5mm | IMRT | -0.007±0.486 | -0.104±0.362 | -0.085±0.24 | -0.078±0.229 |
|  | VMAT | -0.021±0.951 | 0.28±0.785 | 0.332±0.643 | 0.201±0.763 |
|  | P | 0.032 | 0.168 | <0.01 | <0.01 |
| 1.0mm | IMRT | -0.248±1.089 | -0.31±0.521 | -0.352±0.592 | -0.352±0.572 |
|  | VMAT | -0.039±0.898 | -0.056±1.33 | -0.046±0.975 | -0.044±0.901 |
|  | P | 0.034 | 0.174 | <0.01 | <0.01 |
| 1.5mm | IMRT | 0.028±1.367 | -0.464±0.929 | -0.145±0.905 | -0.119±0.926 |
|  | VMAT | 0.323±2.257 | 0.212±1.945 | 0.217±1.69 | 0.065±1.566 |
|  | P | 0.031 | 0.166 | <0.01 | <0.01 |
| 2.0mm | IMRT | -1.662±2.568 | -1.218±1.43 | -1.185±1.393 | -1.173±1.436 |
|  | VMAT | 0.927±3.598 | 1.326±4.052 | 1.84±3.334 | 1.39±3.399 |
|  | P | 0.042 | 0.211 | <0.01 | <0.01 |

**Table 5. Dose sensitivity analysis of MLC error to PTV and OAR in IMRT and VMAT plans.**

| | | Type1 | | Type2 | | Type3 | | Type4 | |
|---|---|---|---|---|---|---|---|---|---|
| | | Dose sensitivity (%/mm) | $R^2$ | Dose sensitivity (%/mm) | $R^2$ | Dose sensitivity (%/mm) | $R^2$ | Dose sensitivity (%/mm) | $R^2$ |
| PTV | IMRT | -0.819 | 0.957 | 1.737 | 0.999 | -2.062 | 0.999 | -0.197 | 0.532 |
| | VMAT | -2.136 | 0.961 | 6.222 | 0.999 | -5.314 | 0.990 | -0.023 | 0.001 |
| Cord | IMRT | 2.568 | 0.775 | 3.286 | 0.999 | -3.718 | 0.999 | -0.616 | 0.498 |
| | VMAT | 0.037 | 0.995 | 6.557 | 0.999 | -5.986 | 0.999 | 0.477 | 0.571 |
| Heart | IMRT | -0.679 | 0.989 | 5.044 | 0.999 | -4.920 | 0.999 | -0.521 | 0.928 |
| | VMAT | -0.697 | 0.999 | 7.786 | 0.999 | -6.236 | 0.990 | 0.491 | 0.580 |
| Lung | IMRT | -1.079 | 0.996 | 4.192 | 0.999 | -4.200 | 0.999 | -0.477 | 0.701 |
| | VMAT | -0.065 | 0.954 | 7.456 | 0.999 | -6.615 | 0.999 | 0.665 | 0.563 |
| Left Lung | IMRT | -0.848 | 0.999 | 4.055 | 0.999 | -4.068 | 0.999 | -0.446 | 0.658 |
| | VMAT | -0.025 | 0.699 | 7.550 | 0.999 | -6.625 | 0.999 | 0.476 | 0.503 |

suggested an action level of 1 mm gap width deviation [16]. Chow et al. examined the VAMT plan for SBRT using log files, and found that the maximum MLC positional error was 0.6 mm [17]. Oliver et al. analyzed the effect of MLC positioning errors for VMAT plans and concluded that the MLC positioning errors for VMAT treatments should be within 0.6 mm to keep the dose variation in the target coverage within 2% [18]. In addition, they reported that the dose sensitivity for systematic MLC gap opening and closing errors in RapidArc plans for prostate cancer was 8.2 and -7.2% per mm, respectively [9]. Ai et al. reported a significant difference in dose sensitivity between MU-weighted and unweighted MLC position errors in SBRT radiotherapy [8]. Thus, MLC errors have a significant impact on the effectiveness of plan execution.

In this study, we evaluated the potential clinical impact of simulated MLC position errors on IMRT/VMAT plans by calculating the gEUD variation in NSCLC SBRT and investigated the difference in dose sensitivity between IMRT and VMAT in the presence of systematic MLC errors and random errors. The results showed that MLC positional errors affected target coverage, OAR protection, CI, and HI of IMRT and VMAT plans [19]. In addition, The IMRT plans are seen to exhibit lower sensitivity than VMAT plans. This was possibly due to the MLC positional errors which depended on gap statistics and jaw sizes. In addition, VMAT requires MLC leaves, gantry positions, and dose rates to change dynamically during irradiation [20,21].

The gEUD was based on both the physical dose information and the radiobiological response of the tumor, whereas the positional accuracy of the MLC leaves can directly cause both radiation volume and dose deviations. Therefore, gEUD was used in this study to evaluate the effects of MLC positional errors on IMRT/VMAT plans [18]. Our data showed that for shift errors, the gEUD sensitivity values of PTV for IMRT and VMAT plans were -0.819 and -2.136%/mm, respectively. The effects are symmetrical with respect to the signs of the systematic errors. It can be seen from Table 5 that the dosimetric effects of systematic errors are strongly linearly correlated with the magnitude of the error. Similar results have been reported previously [9]. The results for systematic opening and closing MLC errors in this study estimated a gEUD sensitivity of -2.0623 and 1.737%/mm for IMRT and -5.3143 and 6.222%/mm for VMAT, which is not consistent with the study by Oliver et al [9]. The explanation for this is, that in SBRT, the target volume is smaller than in conventional irradiation plans; therefore, the subfields formed by the MLC are smaller. The smaller the subfields, the greater the dosimetric impact of opening or closing errors on the treatment plans [18,19]. Fig 5 summarizes the number of control points and the size of the apertures at each control point formed by

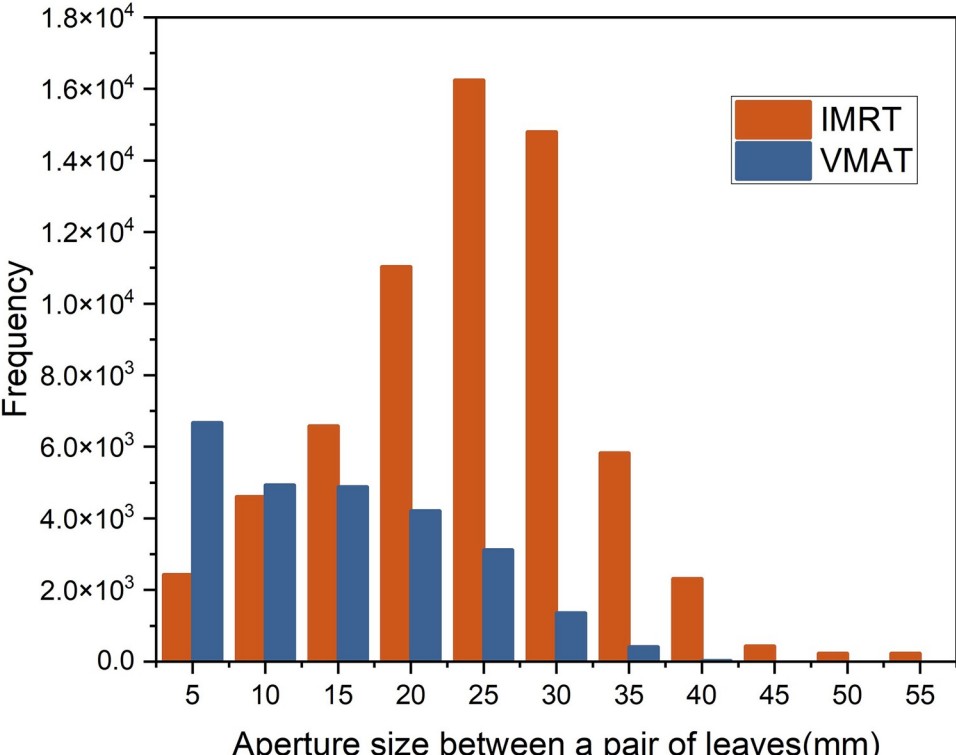

**Fig 5. Frequency histogram of all windows formed by MLC leaf pairs during dynamic window delivery of all IMRT and VMAT plans.**

opposing pairs of MLC leaves during dynamic transmission of the IMRT/VMAT plans. Based on the results shown in Fig 5, it is apparent that IMRT plans require about 13000 different leave pair combinations, of which 21% are separated by less than 2 cm. In contrast, VMAT plans require about 16500 leave pair combinations, of which 64% are separated by less than 2 cm. Further analysis indicated that VMAT tends to use more small-area subfields than IMRT. In particular, VMAT consists of a large number of subfields with stenosis strips, that are much larger than those of IMRT. This is associated with a larger fractional change in relative output factor(ROF) because the ROF is steeper for smaller subfields [18,19]. The mean values for VMAT and IMRT were 2926.31±334.42 and 1971.23±216.45, respectively. Similar to previous studies, dose sensitivity increased with the increasing value of MU [8].

In conclusion, the shift, opening, and closing errors of the MLC leaves had a significant effect on the dose distribution of the SBRT plans, and the effects on VMAT were more pronounced than those of IMRT. The random error had less influence and could be ignored. If a 2% change in gEUD of the PTV was assumed to be an acceptable level of dose deviation due to MLC effects alone, then shift, opening, and closing errors in leaf position for IMRT would have to be limited to 2.4, 0.97, and 1.15 mm, respectively, and for VMAT, they would have to be limited to 0.95, 0.32, and 0.38 mm, respectively [9,22].

The dose sensitivity results obtained in this study can be used to guide clinical quality assurance measures. Systematic MLC errors had a significant impact on the gEUD of SBRT. In particular, VMAT plans have a greater impact on dose distribution than IMRT plans and should be carefully monitored. In this study, we simulated that the MLC errors may be within a certain range and their dosimetric impact on SBRT plans.

## Conclusion

In this study, we investigated the effects of intentional application of MLC position errors on NSCLC SBRT IMRT/VMAT treatment plans. We found that systematic errors had the greatest impact on CI, GI, $D_{2cm}$, and gEUD for all parameters assessed. For VMAT plans, there was a large variability in the MLC errors. Based on the dosimetric results, this study provides a guide for the development of performance standards for multi-leaf collimators.

## Acknowledgments

We would like to thank Xiaobin Chang, Kun Zhang, and Qiang Zhao for their support during the manuscript preparation.

## Author Contributions

**Conceptualization:** Jia Deng, Xiangyang Wu.

**Data curation:** Jia Deng, Yun Huang, Ye Hong.

**Formal analysis:** Jia Deng.

**Funding acquisition:** Jia Deng.

**Investigation:** Jia Deng.

**Project administration:** Jia Deng.

**Resources:** Ye Hong.

**Software:** Jia Deng.

**Validation:** Yun Huang.

**Visualization:** Yun Huang.

**Writing – original draft:** Jia Deng.

**Writing – review & editing:** Jia Deng, Xiangyang Wu, Yaolin Zhao.

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
