## [Decision Letter · Decision Letter 0]

13 Sep 2022

PONE-D-22-19213Comparison of Dosimetric Effects of MLC Positional Errors on VMAT and IMRT plans for SBRT Radiotherapy in Non-small Cell Lung CancerPLOS ONE

Dear Dr. Deng,

Thank you for submitting your manuscript to PLOS ONE. After careful consideration, we feel that it has merit but does not fully meet PLOS ONE’s publication criteria as it currently stands. Therefore, we invite you to submit a revised version of the manuscript that addresses the points raised during the review process.

This study provides further support for an existing body of knowledge. With the changes suggested by the reviewer the manuscript would be significantly strengthened and likely worth publishing.==============================

We look forward to receiving your revised manuscript.

Kind regards,

Devarati Mitra

Academic Editor

PLOS ONE

Journal Requirements:

3. Please note that PLOS ONE has specific guidelines on code sharing for submissions in which author-generated code underpins the findings in the manuscript. In these cases, all author-generated code must be made available without restrictions upon publication of the work. Please review our guidelines at https://journals.plos.org/plosone/s/materials-and-software-sharing#loc-sharing-code and ensure that your code is shared in a way that follows best practice and facilitates reproducibility and reuse. New software must comply with the Open Source Definition.

"JD was supported by grants from the Health research Program of Shaanxi Provincial Health Commission (No. 2022D037)."

Additional Editor Comments:

This study helps provide further support for an existing body of knowledge. As noted by the reviewer while not the most novel concept it is well written and does provide further foundation for current practice. With the changes suggest the study would be much stronger and likely worth publishing.

Reviewers' comments:

Reviewer's Responses to Questions

**Comments to the Author**

1. Is the manuscript technically sound, and do the data support the conclusions?

Reviewer #1: Partly

2. Has the statistical analysis been performed appropriately and rigorously? 

Reviewer #1: Yes

3. Have the authors made all data underlying the findings in their manuscript fully available?

Reviewer #1: Yes

4. Is the manuscript presented in an intelligible fashion and written in standard English?

Reviewer #1: Yes

5. Review Comments to the Author

Reviewer #1: This work is clear and concise with few grammatical errors.

This work is not novel, as other approaches have inserted random and systematic errors into plans to quantify impact on dosimetry. Rather, it reinforces continuing work that monitors performance capabilities of Varian Truebeam MLCs.

General recommendations:

To bring simulated results of this work more into context, I suggest that the authors examine the frequency and magnitude of their own truebeam MLC errors. This would show if experimental results here would actually be of use clinically or for QA purposes. If authors are unable to do so, please consider the citations that are currently in the work about frequency and magnitude of MLC error (Chow et al). When authors recommend that errors must be less than a certain amount in lines 173, this assumes that a systematic shift to all MLCs must exist. Please indicate based on citations or your own clinical values if this situation could occur and/or how it would be controlled or monitored.

Tables in this work do not contain consistent significant digits or units of dose. Please address.

Do clinicians or physicists at your clinic use gEUD to make clinical decisions? How is gEUD a useful metric to monitor for QA? Please address.

Please indicate how random errors were simulated? No information is provided. What distribution was used to create random errors? Please cite the following paper in your discussion, as it also simulates random and systematic errors and investigates dosimetric impact: https://aapm.onlinelibrary.wiley.com/doi/full/10.1002/acm2.12677

Specific comments:

Abstract:

“The aim of this study was to investigate the difference between MLC positional error and dosimetry of volume intensity modulated (VMAT) and general intensity modulated (IMRT) plans for non-small cell lung cancer (NSCLC).”

The usage of the word “difference” is incorrect here and meaning is not correct.

Instead: The aim of this study was to investigate the impact of MLC positional error on dosimetry of volume intensity modulated (VMAT) and general intensity modulated (IMRT) plans for non-small cell lung cancer (NSCLC)

Lines 225

That all leaves would be systematically shifted by these amounts is very unlikely. The Varian Truebeam monitors leaf positions many times over per second and prevents a beam on if leaves are out of tolerance.

Line 227-228

“In particular, VMAT plans have a greater impact on dose distribution than IMRT plans and should be carefully monitored”.

Authors should be more specific. The MLC positions are already closely monitored and recorded in the dynalog files. Please indicate what is meant. How specifically do reviewers plan to use these results to guide clinical QA measures? Do the authors mean IMRTQA? Is this work part of a commissioning process, determination of baseline, or educational in nature?

I am of the opinion that this study is very useful for teaching of medical physicists and demonstrating how different MLC errors can impact a patients treatment (if these errors were to not be caught by MLC control system).

Line 230

“In this study, we simulated that the MLC errors may be within a certain range.”

This sentence is not correct. Instead it should read that “in this study, we simulated MLC errors within a certain range and their dosimetric impact on SBRT plans.”

Line 231

“In future research, it would be useful to conduct experimental studies to determine the typical long term accuracy and precision of the relevant parameters for SBRT delivery”

What relevant parameters do the authors mean? This sentence is not specific. Authors need to please recognize that relevant measures that quantify dosimetric impact of MLC errors is a well-studied sub field of medical physics, so what is the authors motivation for continued monitoring?

6. PLOS authors have the option to publish the peer review history of their article (what does this mean?). If published, this will include your full peer review and any attached files.

Reviewer #1: No

---

## [Author Response · Author response to Decision Letter 0]

20 Oct 2022

Dear Editors:

 Many thanks for your letter and the reviewers’ comments concerning our manuscript entitled “Comparison of Dosimetric Effects of MLC Positional Errors on VMAT and IMRT plans for SBRT Radiotherapy in Non-small Cell Lung Cancer”. These comments are all valuable and very helpful for revising and improving the manuscript. We have revised the manuscript according to your kind advices and the reviewer’s suggestions. The main revisions and responses to the reviewer’s comments are as follows:

Editor:

Journal Requirements:

Reply: Thank you for your suggestion. We have modified it according to the requirements.

Reply: Thank you for your suggestion. We have included an ethics statement in the methods section, as detailed in lines 53-55 of the manuscript. Oral informed consent was obtained by telephone and the transcript has been uploaded in the ethics statement.

3. Please note that PLOS ONE has specific guidelines on code sharing for submissions in which author-generated code underpins the findings in the manuscript. In these cases, all author-generated code must be made available without restrictions upon publication of the work. Please review our guidelines at https://journals.plos.org/plosone/s/materials-and-software-sharing#loc-sharing-code and ensure that your code is shared in a way that follows best practice and facilitates reproducibility and reuse. New software must comply with the Open Source Definition

Reply: Thank you for your suggestion. It was requested that the code be shared. And it can be accessed through the following DOI path: https://doi.org/10.6084/m9.figshare.21342072

"JD was supported by grants from the Health research Program of Shaanxi Provincial Health Commission (No. 2022D037)."

Reply: Thank you for your suggestion. The requested changes were made, and the revised fund statement was incorporated into the cover letter. The content is as follows: “This work was supported by grants from the Health Research Program of Shaanxi Provincial Health Commission (No. 2022D037). There was no additional external funding received for this study.”

Reply: Thank you for your suggestion. 

We have shared our raw data and the minimal dataset at：

https://doi.org/10.6084/m9.figshare.21366744.v1

Our institute's ethics committee can be reached at the following address:

Shaanxi Provincial Cancer Hospital, Xi’an, Shaanxi, People’s Republic of China

Tel：+86-029-85276017

Reply: Thank you for your suggestion. The full ethical statement is included in the methods section, on lines 53-55 of the article.

Reviewer # 1

1. General recommendations:

To bring simulated results of this work more into context, I suggest that the authors examine the frequency and magnitude of their own truebeam MLC errors. This would show if experimental results here would actually be of use clinically or for QA purposes. If authors are unable to do so, please consider the citations that are currently in the work about frequency and magnitude of MLC error (Chow et al). When authors recommend that errors must be less than a certain amount in lines 173, this assumes that a systematic shift to all MLCs must exist. Please indicate based on citations or your own clinical values if this situation could occur and/or how it would be controlled or monitored.

Reply: Thanks for your comments. The purpose of this article is to discuss the sensitivity of MLC errors to PTV and OAR doses in clinical SBRT plans. Our research can help doctors and physicians recognize the relationship between MLC errors and clinical doses. Furthermore, we can provide MLC tolerance specific to SBRT as a treatment reference. According to Oliver et al., there was a linear relationship between various types of MLC errors and PTV gEUD. Table5 shows how the slope parameter from the linear fitting was used to determine the dose (gEUD) sensitivity (%/mm) to MLC position errors. The findings enable us to propose tolerances on MLC leaf positions that are dosimetrically equivalent to those considered acceptable for dose deviations in conventional therapy [2% from TG40 / TG142], at least for the target.

2. Tables in this work do not contain consistent significant digits or units of dose. Please address.

Reply: Thank you for the reminder. We have already corrected. 

In Fig 3, the unit of variation for CI, GI, and D2cm is %, which has been modified in Fig 3.

In Table 1, 2, 3, and 4, the unit of the gEUD variation is %, which has been modified in the Table 1, 2, 3, and 4.

In Table 5, the dose sensitivity is in units of %/mm.

3. Do clinicians or physicists at your clinic use gEUD to make clinical decisions? How is gEUD a useful metric to monitor for QA? Please address.

Reply: Physicists at our clinic use gEUD cost functions for plan optimization, it can be an effective tool for dose sculpting in IMRT/VMAT planning. The generalized EUD can be used to evaluate dose distributions in the tumor and surrounding critical structures. It reports the generalized mean value of the non-uniform dose distribution, which represents the homogenous dose distribution that produces the same local control as that obtained with an inhomogeneous dose distribution for the case of tumors. Moreover, gEUD is closely related to normal tissue complication probability (NTCP). The gEUD is based on the radiobiologic response of the tumor rather than physical dose information, allowing it would be a better predictor of clinical outcome.

The following references provide a more detailed explanation of gEUD. (Niemierko A. Reporting and analyzing dose distributions: a concept of equivalent uniform dose. Med Phys 1997;24(1):103-10. doi:10.1118/1.598063. PMID:9029544). This literature has been cited in the article on lines 96 and 97.

4. Please indicate how random errors were simulated? No information is provided. What distribution was used to create random errors? Please cite the following paper in your discussion, as it also simulates random and systematic errors and investigates dosimetric impact: https://aapm.onlinelibrary.wiley.com/doi/full/10.1002/acm2.12677

Reply: Thanks for your comments. 

In this work, each open leaf position in each control point was modified by a number generated from a Gaussian distribution centered on zero, with a standard deviation equal to the error magnitude. Six magnitudes of random error were simulated. To avoid potential errors and unintentional duplication, we generate a new Gaussian random distribution for each error plan, so that the same errors are not added to different exposure fields and plans.

In line 74 and 75, we have provided additional explanations.

5. Specific comments:

1) Abstract:

“The aim of this study was to investigate the difference between MLC positional error and dosimetry of volume intensity modulated (VMAT) and general intensity modulated (IMRT) plans for non-small cell lung cancer (NSCLC).”

The usage of the word “difference” is incorrect here and meaning is not correct.

Instead: The aim of this study was to investigate the impact of MLC positional error on dosimetry of volume intensity modulated (VMAT) and general intensity modulated (IMRT) plans for non-small cell lung cancer (NSCLC)

Reply： Thank you for the reminder. We have already corrected it.

2) Lines 225

That all leaves would be systematically shifted by these amounts is very unlikely. The Varian Truebeam monitors leaf positions many times over per second and prevents a beam on if leaves are out of tolerance.

Reply： Thanks for your comments. In this work, regarding the dosimetric sensitivity of the PTV and OARs, we implicitly assumed that treatment plans were affected by MLC position alone(line 223). The results were based on this assumption.

3) Line 227-228

“In particular, VMAT plans have a greater impact on dose distribution than IMRT plans and should be carefully monitored”.

The authors should be more specific. The MLC positions are already closely monitored and recorded in the dynalog files. Please indicate what is meant. How specifically do reviewers plan to use these results to guide clinical QA measures? Do the authors mean IMRTQA? Is this work part of a commissioning process, determination of baseline, or educational in nature?

I am of the opinion that this study is very useful for teaching of medical physicists and demonstrating how different MLC errors can impact a patients treatment (if these errors were to not be caught by MLC control system).

Reply：Thanks for your comments. The tolerance of the MLC controller is a user-defined parameter. The optimal tolerance value is tight enough to prevent severe dosimetric changes but clinically practical enough to avoid unnecessary treatment session lengthening due to beam delivery interruptions. The purpose of this work was to evaluate the dosimetric impact of both random and systematic errors in the leaf positions of MLC, rather than to perform a retrospective study of the actual leaf position. TG-142 and ESTRO 2008 Reports recommend defining MLC positioning accuracy with qualitative weekly and quantitative monthly QA. For MLC QA, such as EPID(2D) or ArcCheck(3D) can be used for analysis to quantitatively assess MLC positioning errors, according to the results in this work, a rationalization recommendation can be made for MLC accuracy requirements to provide the accurate delivery of IMRT and VMAT.

4) Line 230

“In this study, we simulated that the MLC errors may be within a certain range.”

This sentence is not correct. Instead it should read that “in this study, we simulated MLC errors within a certain range and their dosimetric impact on SBRT plans.”

Reply: Thank you for the reminder. We have already corrected.

5) Line 231

“In future research, it would be useful to conduct experimental studies to determine the typical long term accuracy and precision of the relevant parameters for SBRT delivery”

What relevant parameters do the authors mean? This sentence is not specific. Authors need to please recognize that relevant measures that quantify dosimetric impact of MLC errors is a well-studied sub field of medical physics, so what is the authors motivation for continued monitoring?

Reply: According to the results of this paper, the dose deviations caused by errors in VMAT plans are greater than those in IMRT plans, so quality control of VMAT has received more attention and monitoring than IMRT. I agree with your assessment and recommendation. The main purpose of this research is to conduct a study of the relationship between MLC errors and clinical geud dosimetry in order to establish clinical criteria for each MLC error tolerance. The previous expression was not entirely accurate. To avoid misunderstanding, this sentence has been removed from the manuscript.

We tried our best to improve the manuscript according to the advices. And we hope that the revision will meet with your approval. Once again, thanks very much for your comments and suggestions. If you have any questions, please contact us at the following address.

Sincerely yours,

Jia Deng

Address: 

Department of Radiation Oncology, Shaanxi Provincial Cancer Hospital, Xi’an, Shaanxi, People’s Republic of China

Tel: +8618192331016

E-mail: dengjia92@yeah.net

---

## [Editor Report · Decision Letter 1]

16 Nov 2022

Comparison of Dosimetric Effects of MLC Positional Errors on VMAT and IMRT plans for SBRT Radiotherapy in Non-small Cell Lung Cancer

PONE-D-22-19213R1

Dear Dr. Deng,

We’re pleased to inform you that your manuscript has been judged scientifically suitable for publication and will be formally accepted for publication once it meets all outstanding technical requirements.

Kind regards,

Devarati Mitra

Academic Editor

PLOS ONE

Additional Editor Comments (optional):

Thank you for addressing the points brought up on review. The manuscript now appears acceptable for publication.
---

## [Editor Report · Acceptance letter]

22 Nov 2022

PONE-D-22-19213R1 

Comparison of Dosimetric Effects of MLC Positional Errors on VMAT and IMRT plans for SBRT Radiotherapy in Non-small Cell Lung Cancer 

Dear Dr. Deng:

I'm pleased to inform you that your manuscript has been deemed suitable for publication in PLOS ONE. Congratulations! Your manuscript is now with our production department. 

Kind regards, 

on behalf of

Dr. Devarati Mitra 

Academic Editor

PLOS ONE